# DNA methylation signatures follow preformed chromatin compartments in cardiac myocytes

Stephan Nothjunge [1,2], Thomas G. Nührenberg[1,3], Björn A. Grüning[4], Stefanie A. Doppler[5], Sebastian Preissl[1,9], Martin Schwaderer[1,2], Carolin Rommel[1,6], Markus Krane[5,7], Lutz Hein[1,8] & Ralf Gilsbach [1]

Storage of chromatin in restricted nuclear space requires dense packing while ensuring DNA accessibility. Thus, different layers of chromatin organization and epigenetic control mechanisms exist. Genome-wide chromatin interaction maps revealed large interaction domains (TADs) and higher order A and B compartments, reflecting active and inactive chromatin, respectively. The mutual dependencies between chromatin organization and patterns of epigenetic marks, including DNA methylation, remain poorly understood. Here, we demonstrate that establishment of A/B compartments precedes and defines DNA methylation signatures during differentiation and maturation of cardiac myocytes. Remarkably, dynamic CpG and non-CpG methylation in cardiac myocytes is confined to A compartments. Furthermore, genetic ablation or reduction of DNA methylation in embryonic stem cells or cardiac myocytes, respectively, does not alter genome-wide chromatin organization. Thus, DNA methylation appears to be established in preformed chromatin compartments and may be dispensable for the formation of higher order chromatin organization.

[1] Institute of Experimental and Clinical Pharmacology and Toxicology, Faculty of Medicine, University of Freiburg, Albertstrasse 25, 79104 Freiburg, Germany. [2] Hermann Staudinger Graduate School, University of Freiburg, Albertstrasse 25, 79104 Freiburg, Germany. [3] University Heart Center Freiburg-Bad Krozingen, Department for Cardiology und Angiology II, Südring 15, 79189 Bad Krozingen, Germany. [4] Bioinformatics Group, Department of Computer Science, University of Freiburg, Georges-Köhler-Allee 106, 79110 Freiburg, Germany. [5] Department of Cardiovascular Surgery, Division of Experimental Surgery, German Heart Center, Lazarettstraße 36, 80636 München, Germany. [6] Faculty of Biology, University of Freiburg, Schänzlestrasse 1, 79104 Freiburg, Germany. [7] DZHK (German Center for Cardiovascular Research)-Partner Site Munich Heart Alliance, Biedersteiner Strasse 29, 80802 München, Germany. [8] BIOSS Centre for Biological Signaling Studies, University of Freiburg, Schänzlestrasse 1, 79104 Freiburg, Germany. [9] Present address: Ludwig Institute for Cancer Research, Gilman Drive 9500, La Jolla, CA 92093, USA. Correspondence and requests for materials should be addressed to R.G. (email: ralf.gilsbach@pharmakol.uni-freiburg.de)

The development of chromosome conformation capture methods such as Hi-C provided insight into spatial chromatin organization[1]. Hi-C data identify different layers of chromatin organization. Topologically associated domains (TADs)[2,3] are one of these layers. TADs represent self-interacting chromatin domains, often separated by genomic insulators like CTCF and are stabilized by the cohesin complex[4]. TADs are thought to act as regulatory units of the genome[5]. A second layer is represented by spatially separated A and B compartments consisting of single or multiple TADs[6]. The spatially segregated A and B compartments have been identified as active and inactive chromatin, respectively[1]. A compartments are enriched for active histone modifications, including H3K27ac, H3K4me1/me3, H3K9me1, the polycomb mark H3K27me3 while B compartments contain the heterochromatin mark H3K9me3[7].

Recent studies suggested an association of DNA methylation with chromatin organization in differentiated cells[8–11] and during early embryogenesis[12]. Previously, DNA methylation has been shown to be crucial for cell development[13]. Especially, reduced CpG methylation at *cis*-regulatory sites is associated with transcription factor occupancy and establishment of cell-type-specific gene expression during cell differentiation[14]. However, the chronology and dependency of DNA methylation and higher order chromatin organization remains unknown.

To study the chronological order of chromatin organization and DNA methylation, we analyze chromatin interactions (in situ Hi-C) and DNA methylation (WGBS) together with gene expression (RNA-seq) during differentiation and maturation of mouse cardiac myocytes (CM), representing a terminally differentiated cell type. To clarify the significance of DNA methylation signatures for chromatin organization, we analyzed embryonic stem (ES) cells lacking DNA methylation[15,16] and adult CM after embryonic ablation of the de novo DNA-methyltransferases 3A and 3B[17,18].

These data show that CM-specific A/B compartments are established during early CM differentiation. These compartments serve as a template for the establishment of partially methylated domains (PMDs) in B compartments and active DNA methylation turnover in A compartments. Additionally, we demonstrate that DNA methylation is dispensable for the establishment of higher order chromatin architecture in ES cells.

## Results

**Chromatin architecture and epigenetic profiling of CM.** Applying in situ Hi-C analysis[7] to pure FACS-sorted CM nuclei[18–20] (Fig. 1; Supplementary Fig. 1, Supplementary Table 1), we identified TADs and A/B compartments of CM. In adult CM, 44.8 % of the genome consists of A compartments (Supplementary Fig. 2a). As anticipated, A compartments of adult CM were enriched for active histone modifications, including H3K27ac, H3K4me1/me3, H3K9me1 as well as CTCF and cohesin (Fig. 1;

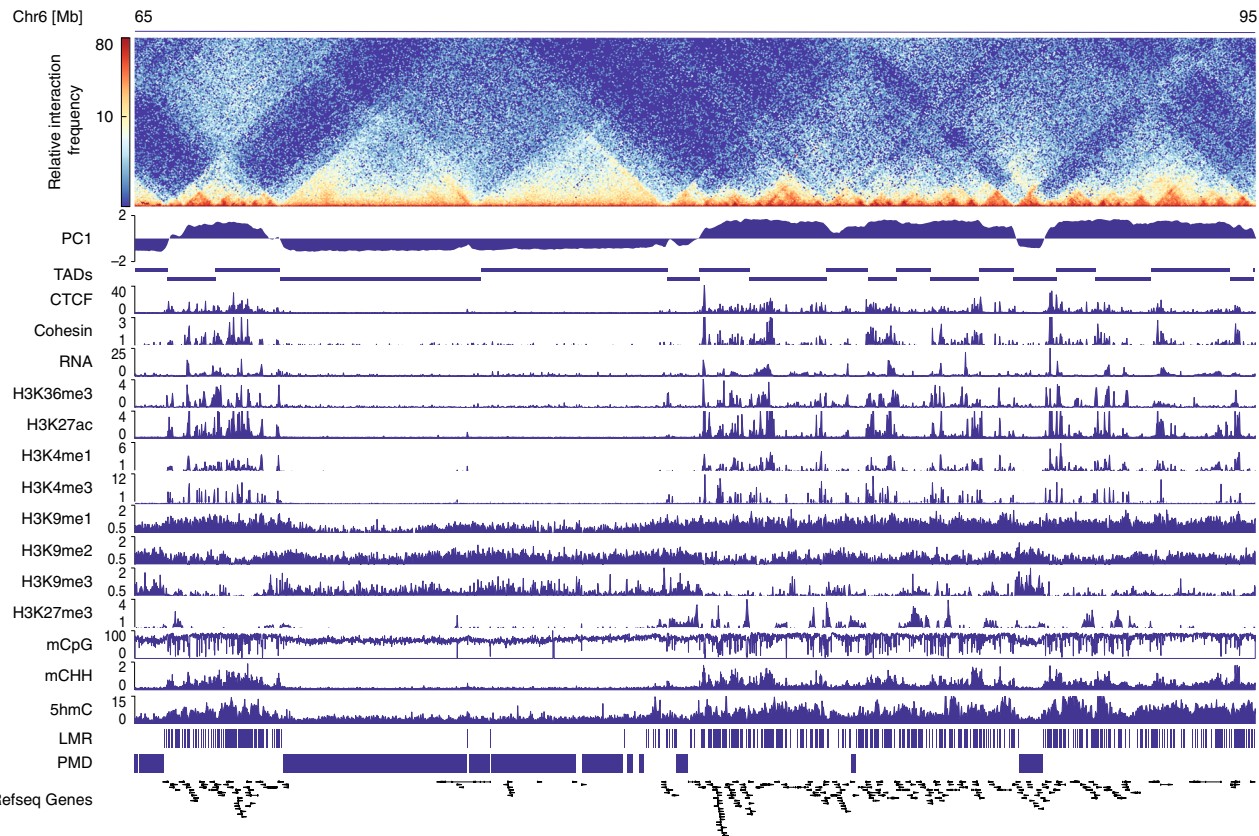

**Fig. 1** Distinct DNA methylation patterns in A and B compartments in adult CM. Hi-C contact maps uncover topologically associated domains (TADs) and multi-TAD A/B compartments in adult CM. Principal component analysis characterizes the A/B status of compartments (A, PC1 > 0; B PC1 < 0). Histone modifications, CTCF, and the cohesion subunit (SMC1) (RPKM) predict the chromatin state. Presence of RNA expression (FPKM) and marking with H3K36me3, H3K27ac, H3K4me1, H3K4me3 classifies A compartments as active. The inactive chromatin status of B compartments is indicated by H3K9me3 enrichment. CpG methylation data (%) shows low-methylated regions (LMRs), characteristic for *cis*-regulatory elements, and partially methylated domains (PMDs) overlapping with A and B compartments, respectively. Non-CpG methylation (mCHH, %), and 5-hydroxymethylcytosine (5hmC, RPKM) mark A compartments. Data shown are from n = 3 Hi-C, n = 3 RNA-seq, n = 3 WGBS; n = 2 5hmC-seq and n = 1–2 ChIP-seq experiments. RPKM reads per kilobase per million, FPKM fragments per kilobase per million mapped reads

Supplementary Fig. 2a). In contrast, B compartments were decorated with the heterochromatin mark H3K9me3 (Fig. 1; Supplementary Fig. 2a). Thus, CM recapitulate the previously described interplay between histone marks and chromatin organization[7].

For the analysis of CpG methylation signatures, we segmented the genome of adult CM into low-methylated regions (LMRs),

representing potential *cis*-regulatory elements such as enhancers (Supplementary Fig. 3)[14], and large PMDs[21] with seemingly disordered CpG methylation. Remarkably, LMRs were characteristic for A compartments and PMDs for B compartments, respectively (Fig. 1; Supplementary Fig. 2a). PMDs have been previously associated with inactive chromatin in other differentiated cell types[9, 11]. Surprisingly, 5-hydroxymethylcytosine

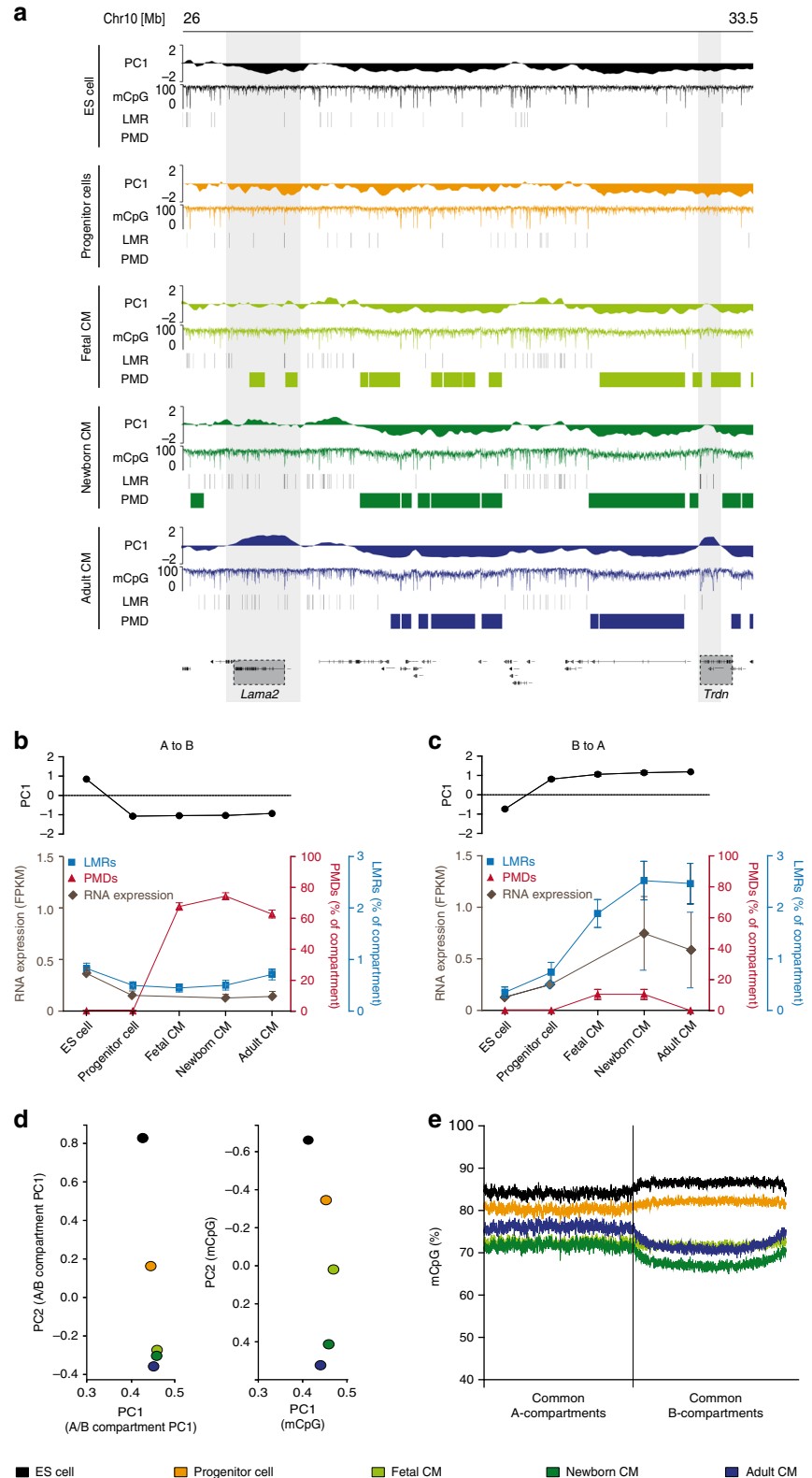

(5hmC) as well as non-CpG (mCHH) methylation clearly demarcated A compartments (Fig. 1; Supplementary Fig. 2a). Thus, A and B compartments show distinct DNA modification signatures including CpG and non-CpG methylation.

**A/B compartments precede manifestation of DNA methylation.** We next analyzed mouse ES cells, cardiac progenitor cells isolated from embryonic hearts[22] (E9-11) and differentiated fetal (E14), newborn (P1), and adult (10–12 weeks) CM to get insights into the dynamics of A/B compartment formation during in vivo CM differentiation and maturation (Supplementary Tables 1 and 2). Compared with ES cells, cardiac progenitor cells and different stages of differentiated CM revealed 7.9% of compartments with dynamic A/B status (Supplementary Figs. 4a and 5a). A compartments which are found in CMs but not in ES cells (CM-A) contain genes essential for CM function, such as myocardin (Myocd) or the sodium–calcium-exchanger Slc8a1 (Supplementary Fig. 5b, c) and display the cell-type-specific character of chromatin compartments. To get insight into the regulatory landscape of A/B compartments in CM, we identified enhancers positive for H3K27ac and H3K4me1 using ChromHMM[23] (Supplementary Fig. 3a). CM-A showed a significantly higher number of enhancers containing motifs characteristic for key cardiac transcription factors, including GATA, MEF2, T-box, and Nkx2, as compared to A compartments shared between ES cells and adult CM (common A; Supplementary Fig. 5e). This suggests that establishment of cell-type-specific compartments like CM-A coincides with recruitment of specific sets of transcription factors.

Visual inspection of a locus with highly dynamic A/B status, containing the laminin subunit alpha2 (Lama2) and triadin (Trdn), as well as genome-wide analysis (Fig. 2a; Supplementary Fig. 4a) imply that compartment switches gradually establish during differentiation. These changes in chromatin organization coincide with the manifestation of DNA methylation signatures and gene expression (Fig. 2a; Supplementary Fig. 4b–d). To analyze the chronology of chromatin organization and DNA methylation, we concentrated on genomic regions showing A/B compartment switches in ES vs. cardiac progenitor cells and CM (Fig. 2b, c). In regions that underwent A to B switches before the CM progenitor stage, PMDs were first apparent in fetal CM (Fig. 2b). Conversely, compartments switching from B to A in ES vs. CM progenitors gained LMRs and gene expression mainly in fetal and newborn CM (Fig. 2c).

We next asked, to which extent genome-wide establishment of A/B compartments and CpG methylation differ during differentiation and maturation. To compare the patterning of A/B compartments between different samples we applied principal component analysis (Fig. 2d; Supplementary Fig. 6). Remarkably, A/B compartment data build a trajectory spanning from ES cells to fetal CM. In contrast, fetal, newborn, and adult CM cluster tightly together, indicating stable A/B patterning in fetal, newborn, and adult CM (Fig. 2d, left panel; Supplementary

Fig. 6). Thus, manifestation of A/B compartments occurs predominantly during differentiation. In contrast, CpG methylation patterns are established gradually during CM differentiation and maturation until the adult stage (Fig. 2d; Supplementary Fig. 6). Taken together our findings clearly show that A/B compartments are mainly established in pluripotent ES cells and multipotent progenitors, whereas CpG methylation is shaped in a continuous process until final maturation to adult CM.

Distinct CpG methylation changes take also place in A/B compartments that are common in ES cells and CM (Fig. 2e). Global analysis of CpG methylation status in common A and B compartments demonstrated that B compartments in undifferentiated ES and progenitor cells are hypermethylated as compared to A compartments. Remarkably, A compartments in differentiated CM showed higher CpG methylation values as compared to B compartments (Fig. 2e). This indicates that hypermethylation of inactive chromatin is restricted to pluripotent and multipotent cells.

**DNMT3 affects CpG methylation and non-CpG methylation in A compartments.** Given these specific changes in DNA methylation, we asked whether modulation of DNA methylation impacts the establishment of A/B patterns during CM maturation (Fig. 3). Since ablation of the maintenance DNA methyltransferase DNMT1 leads to embryonic lethality[24], we first characterized DNA methylation in CM with deletion of the de novo methyltransferases 3a and b (CM-DKO, Dnmt3a$^{-/-}$/Dnmt3b$^{-/-}$)[17, 19]. As expected, we observed a marked CpG hypomethylation upon deletion of DNMT3A/B (Fig. 3a; Supplementary Fig. 7). To our surprise, differentially methylated sites (DMR) were predominantly located in A compartments (Fig. 3a).

This observation also held true in pluripotent human ES cells with ablation of DNMT3A and/or DNMT3B enzymes[25], where hypomethylation took place almost exclusively within A compartments (Supplementary Fig. 8). In good agreement with this observation, enzymes modifying DNA methylation such as TET1, DNMT3A, and B were found to be enriched in A compartments of mouse ES cells (Supplementary Fig. 2b).

In order to comprehensively assess DNA methylation, we also analyzed non-CpG methylation (mCHH) during CM maturation. Similarly to the dynamic changes of CpG methylation, deposition of non-CpG methylation was mainly restricted to A compartments (Fig. 3a). Yet, non-CpG methylation was only detected in adult CM, which are mostly postmitotic (Fig. 3a). Establishment of non-CpG methylation during maturation has previously been detected in neurons[26, 27]. Non-CpG methylation in adult CM is significantly enriched in fully methylated regions (FMR) (Fig. 3c), which mainly correspond to actively transcribed regions marked by H3K36me3 (Supplementary Fig. 2). Ablation of DNMT3A/B prevented the establishment of mCHH in adult CMs (Fig. 3a, d). These findings let us hypothesize that chromatin organization restricts DNA accessibility for DNA modifying enzymes to A

**Fig. 2** DNA methylation and gene expression succeeds formation of A/B compartments. **a** Dynamic A/B status precedes with the formation of LMRs in A compartments and PMDs in B compartments during differentiation and maturation of CM. Depicted is a representative locus harboring the CM-specific genes laminin subunit alpha 2 (Lama2) and triadin (Trdn). **b, c** Analysis of selected compartments with A/B switch manifestation at the progenitor stage (upper panels). B compartments present at the progenitor stage gain PMDs and concordantly loose gene expression after exit from the multipotent stage **b**. Establishment of LMRs and induction of gene expression succeeds formation of A compartments **c. d** Genome-wide principle component analysis (PCA) of A/B compartment values results in a tight cluster of differentiated cardiac myocytes and distant pluripotent ES and multipotent progenitor cells (left graph). Performing PCA analysis of base-pair resolution CpG methylation data results in a trajectory of CM differentiation and maturation with the smallest distance between postnatal stages (right graph). **e** Averaging CpG methylation data for A and B compartment present at all assessed stages (Common-A, Common-B) reveals hypermethylation of B compartments as compared to A compartments in undifferentiated ES and progenitor cells, while differentiated CM show a hypomethylation in B vs. A compartments. Data shown are from $n = 2$-3 Hi-C, $n = 1$-3 RNA-seq and $n = 2$-3 WGBS experiments. Shown are mean ± SEM. FPKM fragments per kilobase per million

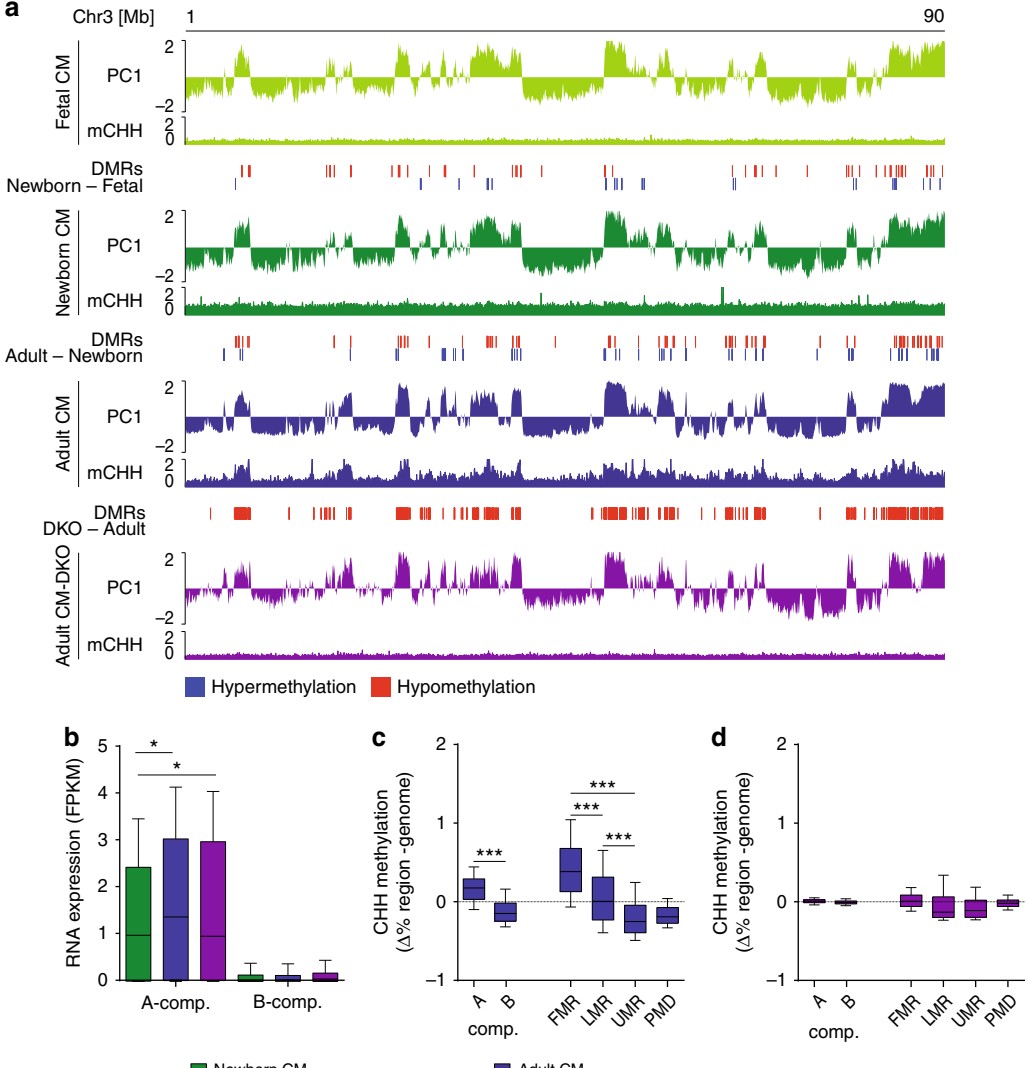

**Fig. 3** DNMT3A/B establishes non-CpG methylation in A compartments of post mitotic CM. **a** Originial traces of PC1 values corresponding to A/B compartments (A, PC1 > 0; B PC1 < 0), regions with differential CpG methylation (Δ > 40%) and CHH methylation (%). Differential CpG methylation during CM maturation and upon fetal ablation of DNMT3 were found in A compartments. CHH methylation is present in A compartments of adult cardiac myocytes in a DNMT3-dependant manner. **b** Gene expression occurs mainly in A compartments of postnatal CM. Ablation of DNMT3A/B had no significant effect on gene expression in CM. **c**, **d** Statistical analysis of CHH-methylation relative to the genome shows significant enrichment in A compartments and in fully methylated regions (mCpG > 85%). No differential CHH methylation is detectable in CM-DKO. Data shown are from $n = 2$–3 Hi-C, $n = 3$ RNA-seq, $n = 2$–3 WGBS experiments. Shown are boxplots with 10–90 percentile whiskers of FPKM values. *$P < 0.05$, ***$P < 0.001$ (ANOVA). FPKM fragments per kilobase per million mapped reads

compartments and thus serves as a barrier for DNA modification dynamics.

**DNA methylation is dispensable for chromatin architecture**. Next, we asked whether modification of DNA methylation affects chromatin structure. Remarkably, ablation of DNMT3A/B in CM had no major impact on A/B patterning, as indicated by highly correlating PC1 values ($r^2 = 0.96$) (Fig. 3a; Supplementary Fig. 9). To test whether DNA methylation is entirely dispensable for the establishment of TADs and A/B structures, we generated Hi-C maps of wild-type (WT) mouse ES cells and ES cells lacking all three DNA-methyltransferases, DNMT1, DNMT3A, and DNMT3B (ESC-TKO[15], Fig. 4a). ESC-TKO are phenotypically not affected by the resulting loss of DNA methylation[15, 28]. Lack of DNA methylation in ESC-TKO did not affect the patterning of A and B compartments (Fig. 4a, b; Supplementary Fig. 7). To confirm this finding, we assessed a second independent ESC-TKO

cell line (cell line 2)[16]. Again, Hi-C experiments did not reveal major changes in the A/B pattern of ESC-TKO compared to WT cells (Supplementary Fig. 10).

Since CTCF binding is DNA methylation-sensitive[14, 29], we tested if the absence of DNA methylation leads to establishment of de novo CTCF binding sites and alters chromatin topology. Notably, Hi-C data of ES-TKOs showed no significant changes in TAD boundaries as compared to ES cells in both tested cell lines (Fig. 4c; Supplementary Fig. 10). Furthermore, analysis of CTCF ChIP-seq data showed highly correlating binding patterns in ES and ES-TKO cells (Fig. 4d).

**Discussion**
Here we report that CM-specific A/B compartments are mainly shaped during early cardiac differentiation. This higher order spatial chromatin structure predefines regions of cell-type-specific DNA methylation signatures. These comprise particularly

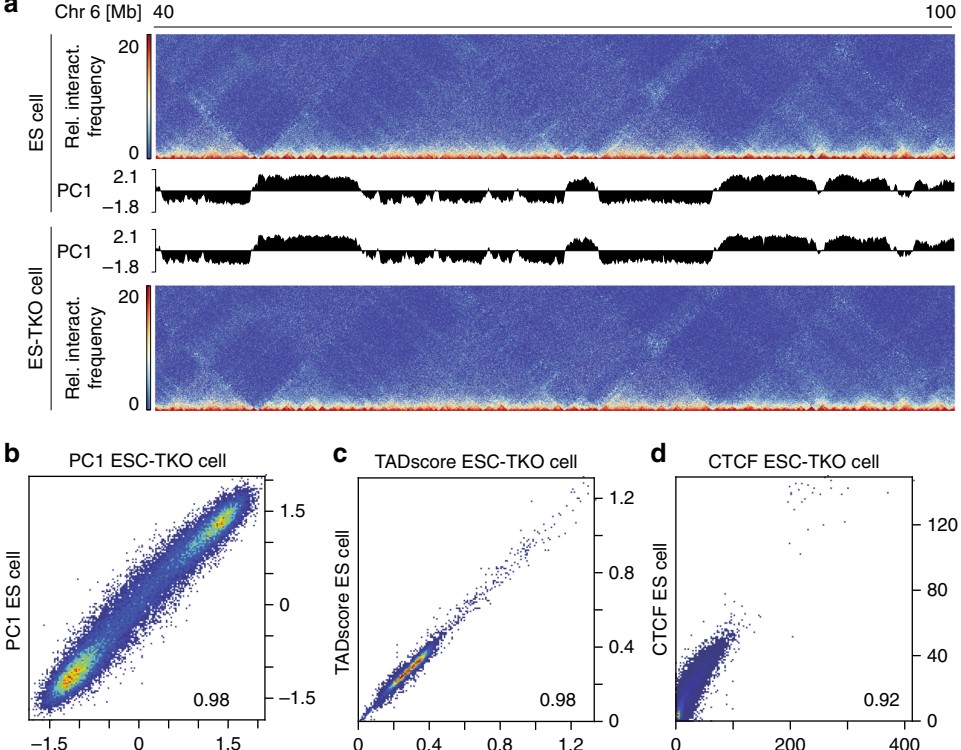

**Fig. 4** Loss of DNA methylation does not alter the chromatin organization in mouse ES cells. **a** Chromatin interaction maps of mouse embryonic stem cells (ES cell line1, upper panel) and ES cells with a complete loss of DNA methylation (ES-TKO cell line 1, *Dnmt1*[−/−]/*Dnmt3a*[−/−]/*Dnmt3b*[−/−]) are indistinguishable. **b**, **c** Genome-wide correlation show that ablation of DNMT-isoenzymes has no effect of A/B pattern (**b** scatter plot and pearson correlation of PC1 values) and insulation of topologically associated domains (**c** TADscore). **d** CTCF peak signals (RPKM) in ES and ES-TKO cells correlates very well suggesting CpG methylation independent CTCF binding. Data shown are merged from $n = 2$ Hi-C experiments

low-methylated regions (LMRs) which represent *cis*-regulatory elements[30] and PMDs[21, 31]. PMDs have been identified previously in various differentiated cell types[31, 32]. Our results show that PMDs establish within preformed B compartments after cell lineage decision in CMs. However, future studies are necessary to unravel the underlying mechanisms and the biological function of PMDs.

Our results clearly show that LMR manifestation is confined to A compartments. These regions were enriched for DNA methylation modifying enzymes, including DNMT3-isoenzymes and TET-isoenzymes. Ablation of the de novo methylation machinery in different cell types induced hypermethylation of A compartments and loss of non-CpG methylation in CM. On the other site 5-hydroxymethylation, which is established by TET-enzymes and indicates active removal of DNA methylation[33] marks A compartments. These observations indicate that accessibility for DNMT3-isoenzymes and TET-isoenzymes is higher in A compartments as compared to B compartments, which is in agreement with a recent publication reporting that A compartments show lower CpG methylation and higher accessibility for DNAse during mammalian embryogenesis as compared to B compartments[12].

Recent studies revealed that the insulator protein CTCF is necessary for proper formation of chromatin loops[34, 35]. A study by Nora et al.[36] used auxin-inducible degradation of CTCF to show that CTCF is essential for the stability of TADs but not for A/B compartments. DNA methylation-sensitive binding of CTCF[29] let us speculate that proper CpG methylation is required for TAD formation. However, our results indicate that loss of DNA methylation in mouse ES cells has no major impact on TAD insulation and does not alter the establishment of CTCF

binding sites in pluripotent cells. Furthermore, a microscopic study found that the dispersed chromatin fiber architecture typical for ES cells is not altered in ES-TKO[37]. These data support our finding that ablation of DNA methylation in pluripotent cells as well as of DNMT3A/B enzymes in CMs has a negligible effect on higher order chromatin organization. In contrast, a targeted methylation of CTCF binding sites by dCas9-Dnmt3a leads to depletion of CTCF binding and disruption of TADs[8]. Probably the genome engineering approach by Liu et al.[8] overrides the physiological DNA demethylation induced by CTCF occupancy[14, 38].

In summary, our findings support a model in which higher order chromatin conformation is a regulatory mechanism guiding cell-type-specific establishment of CpG methylation and non-CpG methylation signatures.

## Methods

**Animal procedures.** Animal procedures were permitted by the responsible Committee on the Ethics of Animal Experiments (Regierungspräsidium Freiburg, Germany and Regierung von Oberbayern, Munich, Germany) and they conformed to the Guide for the Care and Use of Laboratory Animals published by the National Academy of Sciences, 2011. Mice with CM-specific ablation of DNMT3A/B expression were generated by crossing *Dnmt3a*[flox] and *Dnmt3b*[flox] mice with mice expressing cre recombinase under control of the cardiac *Mlc2a* promoter[17, 18]. Cardiac progenitor cells were isolated from mice expressing enhanced green fluorescent protein under control of an Nkx2.5 enhancer (Nkx2.5-enhancer-EGFP)[22]. For all experiments, we used WT mice of the C57BL/6 J strain. Fetal, newborn, and adult hearts were retrieved at embryonic day 14, postnatal day 1, and 8–12 weeks after birth, respectively.

**Origin of embryonic stem cell lines.** In this study, we used two independent ES cell lines with genetic ablation of DNMT1, DNMT3a, and DNMT3b (TKO) and corresponding WT cell lines. WT and TKO cell line 1 was derived from HA36CB1/

159-2 cells[15]. WT and TKO cell line 2 consisted of HA36CB1 and DNMT TKO-133 cells[16].

**Sorting of cardiomyocyte nuclei**. All steps during the isolation, staining and sorting of CM nuclei[18, 19] were performed at ≤4 °C. All buffers contained fresh protease inhibitor (cOmplete Protease Inhibitor Cocktail, Roche) and DTT (1 mM, dithiothreitol). Frozen mouse ventricles were thawed in 3 mL lysis buffer (5 mM CaCl₂, 3 mM MgAc, 2 mM EDTA, 0.5 mM EGTA, 10 mM Tris-HCl, pH 8) and were dissected using Miltenyi gentleMACS dissociator M tubes and the protocol 'protein_01'. An aliquot of 3 mL of lysis buffer supplemented with 0.4% Triton X-100 was added and the suspension was filtered using 40 μm cell strainer (BD Bioscience). After washing the filter with 2 mL lysis buffer, the suspension was centrifuged ($1000 \times g$, 5 min, 4 °C). The pellet was resuspended and overlayed on 1 M sucrose (3 mM MgAc, 10 mM Tris-HCl, pH8), then centrifuged ($1000 \times g$, 5 min, 4 °C) and the pellet resuspended in 500 μL staining buffer (PBS, 5% BSA, 0.2% Igepal CA-630). Nuclei of CMs were stained by rabbit anti-PCM-1 antibody (1:1000, HPA023374, Sigma-Aldrich) and/or mouse anti-PLN antibody (1:1000, A010-14, Badrilla) for 30 min at room temperature (RT) and subsequently with an anti-rabbit secondary antibody conjugated to Alexa 568 (1:1000, A11011, Life Technologies) and/or with anti-mouse secondary antibody conjugated to Alexa 488 (1:1000, A11029, Life Technologies) for 30 min at RT. Then, nuclei were incubated with Draq-7 (1:100, Cell Signaling Technology) for 10 min at RT. CM nuclei were sorted using a S3 cell sorter (BioRad). FSC pulse width was used to exclude doublets from sorting. Sorted CM nuclei were processed immediately.

**Sorting of adult cardiac myocytes for RNA-seq**. Adult mouse hearts were retrograde perfused with digestion buffer (Tyrode's solution, 25 mM butanedione monoxime, 2 mM CaCl2, 0.8 mg/mL collagenase B (Roche, Mannheim, Germany), 0.4 mg/mL hyaluronidase (Sigma), 3 μg/mL trypsin (Sigma)) for 12 min. Single cell suspension were obtained after stopping of the enzymatic digestion (5% FCS) and gentle dissection. After passing through a 100 μm filter CMs were sorted by FACS using a Bio-Rad S3. CMs were identified by high FSC signal and viable cells were discriminated using DRAQ5 (Cell Signaling Technology)[18]. Gates were set to obtain viable CMs.

**Sorting of fetal cardiac myocytes for WGBS experiments**. Fetal CMs (E14) were sorted after enzymatic digestion[18, 19]. Therefore, fetal hearts were dissociated by several rounds of digestion with trypsin (Gibco) in Hank's Balanced Salt Solution (HBSS, Life Technologies). Digestion was stopped by resuspension in HBSS containing 4% FCS. Cells were permeabilized with 0.1% saponin (Sigma) and stained with antibodies against cardiac troponin I (1:1000, ab47003, Abcam) and α-actinin (1:1000, At7811, Sigma) in combination with secondary Alexa 568 and 488 antibodies (1:1000, A11011, A11029, Life Technologies). CMs and non-CMs were visualized using the cell-permeable nucleic acid stain Vybrant DyeCycle Ruby (1:200, V10309, Life Technologies). Sorting of CMs from cardiac cell suspensions was carried out using a BioRad S3 cell sorter. FSC pulse width was used to exclude doublets from sorting.

**Sorting of cardiac progenitor cells**. Cardiac progenitor cells were isolated from embryonic hearts (E9-11) of Nkx2.5-enhancer-EGFP mice[22, 39]. Embryos of the Nkx2.5-enhancer-EGFP mice were collected in E 9.5–11 from timed matings (a positive mating plug indicated E 0.5). For extraction of embryos, mice were anesthetized with isoflurane (2-chloro-2-(difluoromethoxy)-1,1,1-trifluoroethane) and killed by cervical dislocation. Isolated embryos were cut and digested with a collagenase II (10,000 U/mL, Worthington Biochemical Corporation, Lakewood, NJ) and DNase I (10,000 U/μL, Roche) mixture shaking for 1 h at 37 °C to obtain single cell suspension. For embryonic stages >E10 erythrocyte lysis (Red blood cell lysis solution, Miltenyi Biotec, Bergisch-Gladbach, Germany) was additionally performed. Dissociated embryos were washed with PBS and resuspended in PBS/0.5 % BSA/2 mM EDTA for flow cytometry. Dead cells were stained with propidium iodide solution (2 μg/mL, Sigma-Aldrich). GFP-positive cells indicating multipotent cardiac progenitor cells were isolated using a FACS ARIA™ Illu flow cytometer (BD Biosciences, San Jose, CA) and the BD FACSDiva software version 6.1.2 (BD Biosciences). FSC pulse width was used to exclude doublets from sorting. For Hi-C isolated cells were directly flash-frozen in liquid nitrogen. For WGBS and RNA-seq cells were sorted into RLTplus Buffer (Qiagen) containing β-mercaptoethanol (10 μL/mL) to extract DNA and total RNA.

**Whole-genome bisulfite sequencing**. Genomic DNA was extracted (AllPrep DNA/RNA Mini Kit, Qiagen) and DNA-seq libraries were prepared using the NEXTflex Methyl-Seq Library Prep Kit for Illumina (Bioo) using 1–1.5 μg of DNA or in case of the cardiac progenitor cells using the Ovation Ultralow Methyl-Seq DNA Library System (Nugen) using 15–125 ng DNA according to the manufacturer's instructions.

**ChIP-seq**. ChIP-seq was performed from FACS-sorted CM nuclei[18, 19]. The following antibodies were used in this study: CTCF (diagenode, C15410210-50, 4 μg/ChIP), Cohesin (anti-SMC1; biomol, A300-055A, 4 μg/ChIP), H3K9me1 (abcam,

ab8896, 4 μg/ChIP), H3K9me2 (Cell Signaling Technology, #9753, 4 μg/ChIP), H3K9me3 (Diagenode, C15410193, 4 μg/ChIP). For CTCF and Cohesin immunoprecipitation, the iDeal ChIP-seq kit for Transcription Factors (Diagenode) was used with 1 μg of chromatin and for H3K9me1/2/3 the ChIP-IT High Sensitivity Kit (Active Motif) was used with 200 ng of chromatin according to manufacturer's instructions. Sequencing libraries were prepared from the resulting DNA with the NEBNext Ultra DNA Library Prep Kit for Illumina (NEB). To avoid over-amplification of the sequencing library, test qPCR-amplification was carried out to determine minimum cycles for final PCR.

**RNA-seq**. Polyadenylated RNA was isolated from total RNA with magnetic beads (NEBNext Poly(A) mRNA Magnetic Isolation Module, NEB). Libraries were constructed using the NEBNext Ultra RNA Library Prep Kit for Illumina (NEB) according to manufacturer´s instruction.

**In situ Hi-C of FACS-sorted nuclei**. The in situ Hi-C method was adopted from Rao et al.[7] At least 10,000 FACS-sorted cardiomyocyte nuclei (up to 230,000 nuclei per technical replicate) were centrifuged ($1000 \times g$, 5 min, 4 °C), washed with cold PBS and resuspended in PBS. Nuclei then were fixated using fresh PFA (1% final concentration) for 10 min at RT under constant rotation. Fixation was stopped with 2.5 M glycine (final concentration: 0.25 M). Nuclei were centrifuged, washed with PBS ($1000 \times g$, 5 min, 4 °C), and resuspended in 50 μl of 0.5% SDS. Nuclei permeabilization was carried out for 10 min at 62 °C under constant rotation. After adding of 145 μL water and 25 μL 10% Triton × 100, the nuclei were incubated at 37 °C for 15 min. Then, 25 μL of restriction enzyme buffer (NEB buffer 2.1 respectively DpnII-buffer) and 400 U of restriction enzyme (HindIII respectively DpnII, NEB) were added. Restriction was carried out at 37 °C over night under constant rotation. Both, HindIII and DpnII restricted samples were heat inactivated at 62 °C for 10 min. Sticky-ends were filled in by incubation with 1.5 μL of 10 mM dCTP, dGTP, dTTP, and 37.5 μL 0.4 mM biotin-14-dATP (Life Technologies) and 50 U of Klenow-Pol (NEB) at 37 °C for 90 min. Proximity ligation was carried out at RT for 4 h after adding 663 μL water, 120 μL 10× T4 DNA ligase buffer, 100 μL Triton X-100, 12 μL 10 mg/mL BSA, and 2000 U of T4 DNA ligase (NEB). Two DNA precipitation steps were performed after proteinase K decrosslinking (over night) including RNase A digestion (37 °C, 30 min) after first precipitation. The ligated DNA was resuspended in 100 μL water and sheared using 30 cylces of Bioruptur (Diagenode, 30 s, on, 90 s off, 'low'). In total, 200–600 bp fragments were size selected using Ampure Beads XP (Beckman), biotin pull-down was carried out with 100 μL Dynabeads MyOne Streptavidin T1 (Thermo Fisher). Illumina sequencing adapters were ligated to streptavidin bead-bound DNA. To reduce PCR duplicates, a 1:100 dilution of T1 beads (Invitrogen) containing the Hi-C library was taken to determine minimum number of cycles for final library amplification (KAPA SYBR FAST qPCR, Kapabiosystems). The final amplification was performed in technical replicates using undiluted T1 beads and Phusion High-Fidelity DNA polymerase (NEB). PCR products of technical replicates were pooled and cleaned twice using Ampure Beads XP (Beckman, 0.9× beads).

**Hi-C of mouse embryonic stem cells and progenitor cells**. Mouse embryonic mutant (ES-TKO) and WT stem cells (ES-WT) were cultured without feeders on 0.2% gelatine-coated dishes in DMEM with non-essential amino acids, supplemented with 15% fetal calf serum, 2 mM L-glutamine, LIF, and 0.001% β-mercaptoethanol (37 °C, 7% CO₂)[15]. A total of 10⁶ ES cells or 10⁴ FACS-sorted (see above) progenitor cells were used for in situ Hi-C[7]. Briefly, cells were detached and centrifuged ($300 \times g$ for 5 min), then resuspended in 1xPBS ($1 \times 10^6$ cells/mL). Fresh formaldehyde was added to a final concentration of 1% and incubated for 10 min at RT. The crosslinking reaction was quenched by adding glycine (0.25 M final concentration). After subsequent washing, the pellets were centrifuged for 5 min at $300 \times g$ at 4 °C and flash-frozen in liquid nitrogen. Cells were processed directly or stored at −80 °C. Cells were lysed with 300 μL cold lysis buffer (10 mM Tris-HCl pH 8.0, 10 mM NaCl, 0.2% Igepal CA630, freshly added protease inhibitor) for 15 min on ice and then centrifuged at $2500 \times g$ for 5 min. Pelleted nuclei were washed with 500 μL lysis buffer, then permeabilized and processed as described above.

**Sequencing of DNA libraries**. The concentration of DNA libraries was determined by Qubit (Invitrogen) and the insert size using a Bioanalyzer (High Sensitivity, Agilent Technologies). Pooling of multiplexed sequencing samples, clustering and sequencing were carried out as recommended by the manufacturer on Illumina HiSeq 2500 or Nextseq 500. All libraries were sequenced in paired-end mode. Previously published RNA sequencing libraries, constructed using the identical methods applied in this study, were sequenced in paired-end mode to conform to the data generated in this study.

**Analysis of Hi-C data**. All tools used in this study (besides HOMER tools[40]) were implemented into the Galaxy platform[9]. We analyzed only autosomes. Mouse sequencing data were mapped to the reference genome mm9, human ESC data to reference genome hg19.

Raw reads were trimmed as paired-end reads using Trim Galore! (https://github.com/FelixKrueger/TrimGalore) with default parameters. Trimmed reads were mapped as single-end reads to the reference genome mm9 with Bowtie2[41]

using the local alignment mode (--local). Then, properly mapped reads were sorted by their read names (-n) with SAMtools[42].

The HiCExplorer[43] (version 1.7.2, https://github.com/maxplanck-ie/HiCExplorer) was used for processing of mapped reads, normalizing, analyzing, and visualization of Hi-C data.

Using mapped and sorted reads, hicBuildMatrix (-bs 40000 --minMappingQuality = 1) gave the HiC-matrix as output at given resolution.

hicCorrectMatrix was used to determine thresholds for the following correction of the Hi-C matrix. Once set, the matrix was corrected using the command hicCorrectMatrix correct (--filterThreshold –x.x y.y --perchr).

TADs were identified by hicFindTADs. First, the TADscore was calculated using the command hicFindTADs TAD_score (--minDepth 300000 --maxDepth 3000000 --step 300000 -p 50); then the TAD boundaries and TAD domains were called by using hicFindTADs find_TADs (--minBoundaryDistance 400000).

For visualization of the HiC-matrix and the integration of epigenetic data hicPlotTADs was used.

A/B patterns were identified at 40 kb resolution using HOMER tools[40]. A paired-end tag directory of all mapped reads (as described above) was created using the command makeTagDirectory (-removeSelfLigation -removePEbg -genome mm9 -restrictionSite NNNNNN (AAGCTT for HindIII digested and GATC for DpnII digested samples) –removeRestrictionEnds–fragLength 500). This directory was used for further analysis.

runHiCpca.pl (-res 40000 -cpu 50 -genome mm9) gave the A/B pattern of the given Hi-C experiment. The tool automatically creates a background model for normalization of the Hi-C matrix.

Differential A/B pattern were identified by comparing PC1 values per bin. As criteria we defined a different sign and a delta PC1 of 1. For annotation of differential compartments at least two subsequent bins had to fulfill these criteria.

In CM nuclei after ablation of DNMT3A/B enzymes, principle component 2 (PC2) corresponds to the A/B pattern of chromosome 16 from Mb 65 to the end of the chromosome, similar to results described in the initial Hi-C study by Lieberman et al. for chromosome 4 and 5 in a human lymphoblastoid cell line[1]. We therefore exclude this part of Chr 16 from our quantitative analysis, in particular for the calculation of differential A/B pattern and for the correlation of A/B pattern.

To determine the Pearson correlation of the A/B pattern of two or more HiC-matrices, the PC1 values (bedGraph-files) were used as input for multiBigwigSummary (bin size = 40,000 bp, respectively, 500,000 bp for biological replicates). The resulted matrix was visualized using plotCorrelation (default parameters). Both tools are part of deepTools[44].

**Gene ontology analysis.** Genomic bins with cell-type-specific A/B pattern were intersected with Refseq-genes. The obtained genes were used as input for ClueGO v 2.1.6, a Cytoscape plug-in for gene ontology analysis[45] with following settings: Min GO level = 3, Max GO level = 4, GO fusion = true, Kappa score threshold = 0.4, statistical test used = enrichment (right-sided hypergeometric test), correction method used = Bonferroni step down, number of genes = 10, min percentage = 10.

**Analysis of ChIP-seq and 5hmC data.** ChIP-seq[18, 19, 46] raw reads were trimmed and mapped as PE-reads to the reference genome (mm9) using Bowtie2 with default parameters. PCR duplicates were removed by RmDup (part of SAMtools). For data visualization (i.e., heatmaps) we used deepTools[44]. Peaks were called using MACS2 (--qvalue 0.05)[47].

**Analysis of methylation data.** Whole-genome bisulfite sequencing (WGBS) data were mapped using Bismark[48] and DNA methylation data was called using MethylDackel (https://github.com/dpryan79/MethylDackel). Results for both strands were combined after calling of CpG methylation values. Differentially methylated regions were calculated using Methtools (https://github.com/bgruening/methtools)[18]. Regions displaying a mean CpG methylation delta ≥40% over five CpGs in two compared datasets were selected as DMRs. Only CpGs with a minimal coverage of four in both data sets were included in differential methylation analysis. Additional exclusion criteria were a minimal difference of 10% for each individual CpG and a maximal distance of 1 kb between adjacent CpGs. A CpG methylation guided genome segmentation[30] was used to identify LMRs and PMDs. PMDs < 100 kb were excluded, since they mostly represent partially demethylated genic regions of highly expressed genes in CMs. FMRs show an average CpG methylation of at least 85% and were located between LMRs, PMDs and un-methylated regions. After calling of CHH methylation values results from 1 kb bins were combined to increase the coverage.

**Analysis of RNA-seq data.** Quality and adapter trimming of sequencing reads was performed prior to mapping to remove low quality reads and adapter contaminations. RNA-seq data were mapped to the mouse genome (mm9) using STAR[49]. PCR duplicates were removed using SAMtools[42]. Visualizations (i.e., heatmaps) were carried out with deepTools[44].

**Chromatin state analysis and motif analysis.** ChromHMM[23] was used to learn a 10-state model predict chromatin states based on ChIP-seq data. Prediction of

transcription factor-binding sites within strong enhancers were determined using HOMER tools[40]. The motifs used in this study were derived from ChIP-seq data.

**Replicates.** All experiments supporting the main findings of the manuscript were independently replicated. If not stated otherwise, main figures show merged data from all replicates. A detailed overview of biological replicates is given in Supplementary Table 1.

**Data availability.** Hi-C data, DNA methylomes, ChIP-Seq and RNA-seq data supporting the findings of this study have been deposited in the NCBI SRA databases under accession codes PRJNA378914 (Hi-C), PRJNA229470 (WGBS and ChIP-seq), and PRJNA229481 (RNA-seq). Previously published data used for this study are listed in Supplementary Table 2. Additional data that support the findings of this study are available from the corresponding author.

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

## Acknowledgements

We thank Claudia Domisch for excellent technical support. We thank Sebastian Arnold, Dirk Schübeler and Fidel Ramirez for helpful discussions and advice. We acknowledge help from Nils Hammann for isolation of CM from DKO. We thank the Deep Sequencing Facilities at the MPI of Immunobiology and Epigenetics (Freiburg) and EMBL (Heidelberg) for sequencing. We acknowledge the support of the Freiburg Galaxy Team: Prof. Rolf Backofen, Bioinformatics, University of Freiburg, Germany funded by Collaborative Research Centre 992 Medical Epigenetics (DFG grant SFB 992/2 2016) and German Federal Ministry of Education and Research (BMBF grant 031 A538A RBC (de. NBI)). We are grateful for mouse ES cells and ES-TKO cells from Dirk Schübeler. This study was supported by the Deutsche Forschungsgemeinschaft SFB 992 project B03, DFG projects GI 747/2-1, PR 1668/1-1 and HE 2073/5-1, the BIOSS Centre for Biological Signalling Studies, the Innovationsfonds Baden-Württemberg and DZHK B 15-005.

## Author contributions

R.G. designed the project. S.N. and R.G. performed Hi-C and ChIP-seq experiments. S.P., R.G., and T.G.N. performed WGBS-seq. S.P., R.G., T.G.N., and S.N. performed RNA-seq. M.S. performed ChromHMM annotation. C.R. performed CTCF ChIP-seq experiments. S.A.D. isolated cardiac progenitor cells. R.G. and S.N. analyzed and visualized the data. B.A.G. developed bioinformatic tools. S.N., L.H., and R.G. wrote the manuscript. T.G.N., S.A.D., and M.K. edited the manuscript. All authors discussed the results and commented on the manuscript.

## Additional information

**Competing interests:** The authors declare no competing financial interests.

