## [Peer Review File · Nature Communications]

Reviewers' comments:

Reviewer #1 (Remarks to the Author):

The authors perform an extensive set of experiments to catalog 3D organization of the genome during in vivo and in vitro cardiac differentiation, and correlate A/B compartments with DNA methylation. The data will be a useful resource. However it is unclear to me what the significant advance is, and the negative results are unfortunately not particularly informative.

One concern regards experimental design. If I understand correctly, data are an aggregate of adult CMs, fetal CMs, and external datasets. The lack of consistency in the selection of disparate datasets is quite concerning, as are the low n (sometimes single sample) for many.

Minor point: in the following sentences, "Recent studies revealed that A/B patterning can be established independently of the formation of TADs and associated loops²⁵. CTCF is the main insulator protein in mammals and necessary for proper formation of chromatin loops²⁶." An important recent reference was omitted: Nora et al Cell 2017, which shows conclusive evidence for both statements. Reference 26 is not applicable to the function of CTCF. Initial reference to TADs should not only include Dixon et al 2012 but also Nora et al 2012, published alongside ref 2 in the same journal.

Reviewer #2 (Remarks to the Author):

In this manuscript, Nothjunge and colleagues probe chromatin state during various stages of murine cardiomyocyte development. The developmental time points range from the ES cell stage to adult cardiomyocytes. Assays performed include Hi-C, whole genome bisulfite sequencing, DNA 5-hydroxymethyl-seq, RNA-seq and ChIP-seq for various histone marks (e.g. H3K27ac, H3K9me3) and chromatin-associated proteins (e.g. CTCF, Cohesin complex). Based on these chromatin state maps, they are able to define TADs and multi-TAD A/B compartments for each stage. They found that A-compartments are enriched for low methylated regions (LMRs) while B-compartment are enriched for partially methylated regions (PMRs). They observe that the establishment of A/B compartmentalization largely occurs during very early stages of cardiomyocyte lineage commitment, with no major detectable changes in later stages of cardiomyocyte maturation. In contrast, CpG methylation occurred in a more gradual and continuous process throughout the time points studied, including the later stages of postnatal maturation. Differential CpG methylation occurred mainly in A-compartments. The authors then asked whether genetically eliminating dynamic DNA methylation (using ES cells null for Dnmt3a, 3b and 1) would alter A/B compartmentalization. Interestingly, lack of DNA methylation in these triple KO mouse ES cells did not alter the patterning of A/B-compartments, as detected by HiC. The authors conclude that A/B-compartmentalization is largely established during very early stages of cardiomyocyte lineage commitment and that total loss of DNA methylation does not affect the architecture of A/B compartments in mouse ES cells.

This is a very well-executed and extensive chromatin profiling study that uses many complementary methods to probe chromatin state. The experiments are technically very challenging, especially those that probe cells in the later stages of primary cardiomyocyte development and maturation. Their technique using sorted cardiomyocyte nuclei for downstream interrogations provides good signal resolution and throughput (i.e., ability to perform multiple ChIPs from a sample). Furthermore, the computational analyses are well executed. As such, the authors have provided an important technical resource for the cardiac field. In addition, their studies of cells deficient in DNA methylation provide some important and intriguing insights into the temporal dependency of A/B-compartmentalization vs. dynamic DNA methylation.

This reviewer has two major critiques:

1) It is not explicitly clear whether the authors have performed independent biological replicates of the experiments. If the N=1, the authors must explicitly state this in the results and methods. It would be nice if the key findings of the study are reproduced in an independent biological replicate. Given the extensive nature of the chromatin profiles performed here, it is not essential to fully repeat every single experiment. However, replicating the central claims of the manuscript might be useful. These central claims include (a) the temporal dynamics of A/B-compartmentalization vs. dynamic CpG methylation, and (b) the observation that triple Dnmt KO mouse ES cells maintain ability to form A/B-compartments.

2) A recent paper by Nora and colleagues (Nora EP et. al, Cell 2017; PMID 28525758) used a genetically encoded degron system in mouse ES cells to reversibly deplete CTCF and demonstrated a critical role for CTCF in TAD formation and organization. Surprisingly, key features of active vs. inactive genomic compartments remained preserved during CTCF depletion. The authors should briefly discuss their current findings pertaining to A/B-compartmentalization and dynamic DNA-methylation in context of the data in this published paper.

Response to Reviewers:

1) Significance of findings

Reviewer 1: However it is unclear to me what the significant advance is, and the negative results are unfortunately not particularly informative.

Response: The interplay of chromatin organization and epigenetic marks is of great interest and remains mainly unresolved. Especially the relationship between DNA methylation and chromatin architecture has been proposed and discussed in recent studies (Ke et al. Cell 170: 367-381, 2017; Liu et al., Cell 167: 233-247, 2016). The present manuscript clearly shows the chromatin architecture-dependent establishment of DNA methylation patterns. Furthermore, it shows for the first time that DNA methylation is dispensable for chromatin organization in ES cells. These insights are novel and should be of great interest for the scientific community.

2) Number of replicates

Reviewer 1: If I understand correctly, data are an aggregate of adult CMs, fetal CMs, and external datasets. The lack of consistency in the selection of disparate datasets is quite concerning.

Response: All displayed cardiac myocyte and progenitor cell data have been generated by our group using the same methods. Part of the WGBS and ChIP-seq data have been published previously by our group (Gilsbach et al. 2014, Nat. Commun., 5, 5288, 1-13; Preissl et al. 2015, Circ. Res., 117, 413-23) and was therefore listed in Supplementary Table 2 together with data of external groups. In the revised version, we discriminate clearly between data generated by our and external groups (Supplementary Tables 1 and 2).

Reviewer 1: The low n (sometimes single sample) for many is concerning.

Reviewer 2: It is not explicitly clear whether the authors have performed independent biological replicates of the experiments. If the N=1, the authors must explicitly state this in the results and methods. It would be nice if the key findings of the study are reproduced in an independent biological replicate. Given the extensive nature of the chromatin profiles performed here, it is not essential to fully repeat every single experiment. However, replicating the central claims of the manuscript might be useful. These central claims include (a) the temporal dynamics of A/B-compartmentalization vs. dynamic CpG methylation, and (b) the observation that triple Dnmt KO mouse ES cells maintain ability to form A/B-compartments. (Reviewer 2)

Response: We have generated in situ Hi-C, DNA-methylation and RNA-seq data from at least two biological replicates for all assessed samples (Supplementary Table 1 and 2).

In our revised manuscript, we visualized the results of the independent biological replicates to support our main findings as suggested by Reviewer 2 (see Supplementary Figure 1 and 6).

Due to the unexpected observation that triple DNMT-TKO mouse ES cells maintain TADs and A/B-compartments when compared with wild-type ES cells, we already replicated these experiments using a second independent DNMT-TKO cell line (Tsumura et al., 2006). We added these new data to the manuscript (see Supplementary Figure 10).

Therefore, we generated the following new supplementary figures:

1) A correlation plot of A/B-compartment values (PC1) obtained from Hi-C data of independent biological replicates (Supplementary Figure 1).

2) A replication of Figure 2d showing the results from individual biological replicates illustrating the temporal dynamics of compartments vs. CpG methylation (Supplementary Figure 6).

3) Figure 4 of our manuscript shows Hi-C data obtained from a DNMT-TKO ES cell line generated by the Schübeler lab using CRISPR-Cas9 (Domcke et al. 2015, Nature, 528:575–579) and the corresponding wild type ES cell line (HA36CB1/159-2). The new figure (Supplementary Figure 10) shows the results of a second independent DNMT-TKO (TKO-133) and corresponding wild type (HA36CB1) mouse ES cell line generated by the Okano lab (Tsumura et al., 2006, Genes Cells. 11(7):805-14).

Currently, we do not have replicates for H3K9me1, H3K9me2, H3K9me3, CTCF and cohesin. In line with reviewer 2, we think that these experiments are not essential for the main conclusions of our manuscript.

3) Additional references

Reviewer 1: An important recent reference was omitted: Nora et al Cell 2017, which shows conclusive evidence for both statements. Reference 26 is not applicable to the function of CTCF. Initial reference to TADs should not only include Dixon et al 2012 but also Nora et al 2012, published alongside ref 2 in the same journal.

Reviewer 2: A recent paper by Nora and colleagues (Nora EP et. al, Cell 2017; PMID 28525758) used a genetically encoded degron system in mouse ES cells to reversibly deplete CTCF and demonstrated a critical role for CTCF in TAD formation and organization. Surprisingly, key features of active vs. inactive genomic compartments remained preserved during CTCF depletion. The authors should briefly discuss their current findings pertaining to A/B-compartmentalization and dynamic DNA-methylation in context of the data in this published paper.

Response: *We cited and discussed the mentioned and important new study by Nora et al. 2017. In addition, we added the suggested citation for the discovery of TADs (Nora et al. 2012). We think that Reference 26 (Sanborn et al. 2015, PNAS) is correct. The authors describe the CTCF- and cohesion-dependent extrusion model and show that ablation of CTCF binding sites by CRIPR alters loop formation. In the revised version, we cite in addition Fudenberg et al. 2016 (Cell Rep).*

REVIEWERS' COMMENTS:

Reviewer #1 (Remarks to the Author):

The authors have somewhat improved their manuscript with additional data. My opinion of the value of the work to the general community still stands, and the disparate sources of data remain a concern.

Reviewer #2 (Remarks to the Author):

The authors have been responsive to this reviewer's comments and critiques. There are no further comments at this stage.

Response to reviewers comments:

Reviewer #1 (Remarks to the Author):

The authors have somewhat improved their manuscript with additional data. My opinion of the value of the work to the general community still stands, and the disparate sources of data remain a concern.

Response: The findings of this manuscript show the dependency of DNA methylation and chromatin organization on a genome-wide scale. This interplay explains the formation of DNA methylation signatures and regions of active DNA methylation turnover in a chromatin organization-dependent manner. This information is not only of importance for the cardiac community but also for researchers interested in epigenetic mechanisms.

To obtain consistent data we generated all cardiac myocyte data in our lab using identical protocols. This applies to data previously published by our group and data generated for this manuscript (Supplementary Table 1). All Hi-C data of ES- and TKO-cells were generated for this manuscript in our lab, too (Supplementary Table 1).

Publically available WGBS and ChIP-seq data from ES-cells were reanalyzed to support the findings of this manuscript (Supplementary Table 2).

Thank you very much for reviewing our manuscript.

Reviewer #2 (Remarks to the Author):

The authors have been responsive to this reviewer's comments and critiques. There are no further comments at this stage.

Response: Thank you very much for reviewing our manuscript.